# *RASAL1* and *ROS1* Gene Variants in Hereditary Breast Cancer

**DOI:** 10.3390/cancers12092539

**Published:** 2020-09-07

**Authors:** Federica Isidori, Isotta Bozzarelli, Simona Ferrari, Lea Godino, Giovanni Innella, Daniela Turchetti, Elena Bonora

**Affiliations:** 1Genetic Laboratory, Department of Medical and Surgical Sciences, University of Bologna, 40138 Bologna, Italy; federica.isidori2@unibo.it (F.I.); isotta.bozzarelli2@unibo.it (I.B.); giovanni.innella@studio.unibo.it (G.I.); 2Unit of Medical Genetics, Azienda Ospedaliero-Universitaria Policlinico S. Orsola-Malpighi, 40138 Bologna, Italy; simo.ferrari@unibo.it (S.F.); lea.godino@aosp.bo.it (L.G.)

**Keywords:** breast cancer, whole exome sequencing, *RASAL1*, *ROS1*

## Abstract

**Simple Summary:**

Breast cancer is the second leading cause of death in women. Identifying novel genetic factors conferring BC predisposition is crucial to predict who is at increased risk of developing the disease, allowing for early detection and therapy, and optimized patient management. We identified germline pathogenic variants in familial breast cancer patients in *ROS1* and *RASAL1* genes. Further analysis in independent patient group will help understanding the role of these novel genes in breast cancer predisposition.

**Abstract:**

Breast cancer (BC) is the second leading cause of death in women. BC patients with family history or clinical features suggestive of inherited predisposition are candidate to genetic testing to determine whether a hereditary cancer syndrome is present. We aimed to identify new predisposing variants in familial BC patients using next-generation sequencing approaches. We performed whole exome sequencing (WES) in first-degree cousin pairs affected by hereditary BC negative at the *BRCA1/2* (BReast CAncer gene 1/2) testing. Targeted analysis, for the genes resulting mutated via WES, was performed in additional 131 independent patients with a suspected hereditary predisposition (negative at the *BRCA1/2* testing). We retrieved sequencing data for the mutated genes from WES of 197 Italian unrelated controls to perform a case-controls collapsing analysis. We found damaging variants in *NPL* (N-Acetylneuraminate Pyruvate Lyase), *POLN* (DNA Polymerase Nu), *RASAL1* (RAS Protein Activator Like 1) and *ROS1* (ROS Proto-Oncogene 1, Receptor Tyrosine Kinase), shared by the corresponding cousin pairs. We demonstrated that the splice site alterations identified in *NPL* and *ROS1* (in two different pairs, respectively) impaired the formation of the correct transcripts. Target analysis in additional patients identified novel and rare damaging variants in *RASAL1* and *ROS1*, with a significant allele frequency increase in cases. Moreover, *ROS1* achieved a significantly higher proportion of variants among cases in comparison to our internal control database of Italian subjects (*p* = 0.0401). Our findings indicate that germline variants in *ROS1* and *RASAL1* might confer susceptibility to BC.

## 1. Introduction

Breast cancer (BC) is the most common cancer in women worldwide and the second leading cause of cancer death, exceeded only by lung cancer [1].

BC patients with a family history or clinical features (such as bilateral, or early-onset BC) suggestive of inherited predisposition are candidate for genetic testing to determine whether a hereditary cancer syndrome is present. The two major genes *BRCA1* (BReast CAncer gene 1) and *BRCA2* (BReast CAncer gene 2), together with rare high penetrance genes (*TP53* (Tumor protein 53), *PTEN* (Phosphatase and tensin homolog), *CDH1* (Cadherin-1) and *STK11* (Serine/threonine kinase 11)) and moderate penetrance genes collectively explain ~30% of familial BC risk [2]. 

Increasing the knowledge on genetic predisposition is crucial to extend the benefits of targeted surveillance/prevention to still unidentified high-risk women. As an example, among *BRCA1* mutation carriers, the estimated lifetime risk of BC is 67%, and lifetime risk of ovarian cancer is 45%. *BRCA2* mutation carriers face a similar risk for BC, with an ovarian cancer risk estimated to be 12% [3,4]. Therefore, lifetime risk is dramatically increased if compared to the general population, in which BC risk is approximately 13% and the risk of ovarian cancer is 1.5%, with significant implications for prevention and early detection [3,4]. In recent years, large genome-wide association studies (GWAS) have discovered many additional low-penetrance common variants associated with BC risk, with more than 100 SNPs (single nucleotide polymorphisms) independently associated with BC risk [5,6,7]. A recent association study in women of European origin focused on ER-negative disease, or *BRCA1* mutation carriers, who are more likely to develop ER-negative disease (70–80% of cases) [5,8]. The Authors identified independent associations at genome-wide significance level with 10 variants at 9 novel loci. Replicated association was also detected for 10 of 11 variants previously reported in ER-negative disease or *BRCA1* mutation carrier GWAS and consistent association with ER-negative disease was also detected for 105 susceptibility variants identified by other studies [5]. Together, these 125 variants explained ~14% of an assumed twofold increased risk of developing ER-negative disease for the first-degree female relatives of women affected with this subtype and ~40% of the estimated familial risk. An additional study identified 65 novel loci for breast cancer susceptibility [6]. Collectively, the >180 identified low-risk loci can explain 18% of the familial risk. Further studies of the altered molecular pathways are demanded in order to translate the association data into actionable targets for prevention/treatment of the disease. 

In parallel, the advent of next-generation sequencing (NGS) has led to a timely testing of multiple genes, many of which with differing estimated cancer risks. This expansion introduced significant issues in clinical settings, most notably who to test and which genes should be included, since for multiple genes there is little clinical information available to guide patients if a pathogenic variant is identified, and larger panels have a higher incidence of finding variants of unknown significance [9]. Affordable wholeexome sequencing (WES) has become a common approach to identify rare variants, by performing a staged study starting with sequencing in a small cohort of cases with familial aggregation of BC, assuming that genetic contribution is higher. This approach led to the recent identification of rare risk variants in *REQL* (RecQ-like) and *BLM* (BLM RecQ Like Helicase) genes [10,11]. 

We undertook a WES-based approach in affected first-degree cousins, to identify candidate gene variants for hereditary BC (HBC), and identified damaging variants in *NPL* (N-Acetylneuraminate Pyruvate Lyase), *POLN* (DNA Polymerase Nu), *RASAL1* (RAS Protein Activator Like 1) and *ROS1* (ROS Proto-Oncogene 1, Receptor Tyrosine Kinase)genes. We then performed a targeted NGS analysis of these genes identifying additional pathogenic variants in *ROS1* and *RASAL1* in an independent cohort of 131 familial BC patients.

## 2. Results

### 2.1. Whole Exome Sequencing (WES) Analysis

To identify novel predisposing BC loci, we carried out a WES analysis in affected first-degree cousin-pairs with familial BC, who tested negative for BRCA1/2. Since first-degree cousins, based on their relationship, share 1/8 of the entire genome, this strategy helped filtering and selecting shared candidate variants, according to an autosomal dominant model of inheritance. 

Three cousin-pairs were analyzed through WES and candidate variants were identified in different genes (Figure 1a–c; Table 1). 

Variant prioritization for shared heterozygous variants was performed considering (i) a minor allele frequency (MAF) <0.01 in the Genome Aggregation database, gnomAD [12], and an in-house database of 650 Italian exomes; (ii) the pathogenicity of the variants (predicted by the Mendelian Clinically Applicable Pathogenicity (M-CAP) Score [13] and PolyPhen-2 [14] programs for missense variants, and Human Splice Finder v3.1 (HSF v3.1) [15], for the analysis of potential splicing alterations). This led to the identification of the following variants shared by the corresponding couples of cousins (Table 1): a single nucleotide variant abolishing the canonical donor splice site in intron 37 of the gene ROS1 (c.6005 + 2T > G) in family ITA_BC01; two rare variants in family ITA_BC02; a missense change in RASAL1 (p.Phe602Leu), present in ClinVar with conflicting data on pathogenicity and a predicted damaging missense change in POLN (p.Asp229Ala); a splice site deletion in NPL (c.68 + 2delT) in family ITA_BC03. All variants were confirmed by direct sequencing. 

As variants in these genes are not known to be associated to HBC, to explore their role in BC, we investigated whether they are de-regulated in BC, compared to normal tissues. To this aim, we first evaluated the expression profiles of NPL, POLN, RASAL1 and ROS1 in human normal tissues, as reported in GTEx [16] and Human Protein Atlas [17] databases. NPL, POLN and RASAL1 showed a widespread expression in many human tissues, whereas ROS1 showed a more restricted expression in normal conditions, mainly in lung and brain (Appendix A). BRCA1 and BRCA2 also showed a variable degree of expression in human normal tissues (Appendix A).

As reported in the Catalogue of Somatic Mutation in Cancer (COSMIC) [18] database, NPL was overexpressed in 1.18% of BC cases (13/1104), POLN in 3.99% (44/1104), RASAL1 in 6.97% (77/1104) and ROS1 in 1.36% (15/1104). In comparison, overexpression of BRCA1 was observed in 11.02% of cases (122/1104) and overexpression of BRCA2 in 9.42% of cases (104/1104). 

In the Human Protein Atlas [17] NPL, POLN and RASAL1 showed generally an increased expression in most cancer tissues (Appendix A). In particular, RASAL1 overexpression resulted significantly correlated with a worse outcome (increased death) in endometrial cancer (Appendix A), whereas NPL overexpression was correlated to a worse outcome in liver and renal cancers (Appendix A). However, in other studies RASAL1 was found significantly downregulated in cancer [19,20], whereas its overexpression blocked cancer progression [21]. 

### 2.2. Splice Site Alterations in NPL and ROS1

As two of the variants identified via WES affected canonical splice sites, in NPL and ROS1, we evaluated their effect on the final transcripts. The NPL variant altered the canonic splice site in intron 2, through a deletion of the T in position + 2 (c.68 + 2delT). Since no fresh RNA was available from patients’ tissues, we used an in vitro system to study the splicing event in the presence of the variant. We inserted the wild-type or mutant genomic region surrounding exon 2 into the minigene plasmid P1-Altermax, that carries synthetic exons separated by an intronic region where the multiple cloning site maps (Figure 2a). The vectors were transiently transfected into simian COS7 cells, because they have a splicing apparatus comparable to the human one [22]. After RNA extraction and Reverse Transcriptase-PCR (RT-PCR) with primers specific for the synthetic exons, PCR products were sequenced (Figure 2b,c). In presence of the mutant allele, NPL exon 2, which contains the ATG start codon, was not included in the final transcript, suggesting that the protein would not be correctly translated (Figure 2c).

The ROS1 variants identified via WES affected a canonic donor splice site through the substitution in position + 2T > G in the intron 37 of ROS1. Since no fresh RNA was available from the patients’ tissues, we used the in vitro system to study the splicing events in the presence of the mutation. We inserted the wild-type or mutant genomic region surrounding exon 37 into the minigene plasmid P1-Altermax (Figure 3a) and the vectors were transiently transfected into simian COS7 cells. After RNA extraction and RT-PCR with primers specific for the synthetic exons, PCR products were sequenced (Figure 3b,c) in the presence of ROS1 mutant plasmids the splicing machinery used a cryptic intronic splice site 31 bp downstream of the canonical one, leading to the insertion of a premature stop-codon in the protein product of the mutant ROS1 transcript (Figure 3c). 

### 2.3. Variant Screening in BC Patients Negative at BRCA1/BRCA2 Testing

To evaluate the frequency of damaging/possibly pathogenic variants in NPL, POLN, RASAL1 and ROS1 in familial BC, we used a target panel covering all exons and exon-intron boundaries for these genes (custom-made TruSeq amplicon kit, designed with the DesignStudio software [23]). The variant screening was performed in 131 independent BC patients who had previously tested negative at BRCA1 and BRCA2 analysis (Appendix A). The filtering criteria were the same used for the WES analysis. In POLN we identified two rare variants and 1 in NPL in non-coding regions (Appendix A), therefore ruling out a major role of these genes in predisposing to BC in this group of patients.

In the gene ROS1 we identified a total of 15 different heterozygous variants, eight of which showed a significant increased allele frequency in our sample vs. gnomAD [12] database (European non-Finnish population used as controls) and two were novel variants, p.Ile1305Phe and p.Gly1991Arg (Table 2). Several variants recurred among patients, such as (i) the c.-32C > T variant in the 5′UTR region, present in two unrelated HBC individuals and absent in the control European group; (ii) the missense variant p.Glu1902Lys, present in three unrelated patients, and showing an increased frequency compared to the controls (*p* = 0.0024; Fisher’s exact test, Table 2); (iii) the same splice site variant c.6005 + 2T > G, identified via WES was detected in another unrelated patient (BC14177); (iv) the missense variant p.Arg2126Gln, present in three unrelated patients, showing a significant difference in allele distribution compared to gnomAD [12] (*p* = 0.0008; Table 2). 

Rare/novel heterozygous damaging variants were also identified in RASAL1 including: (i) the premature stop codon c.158G > A p.Trp53Ter (found in cis with the novel missense change p.Ile83Asn); (ii) a novel frameshift p.Leu292CysfsTer5 in another patient; (iii) the novel missense changes, predicted damaging, p.Ala43Thr and p.Ser466Asn in two unrelated patients (Table 3), all pointing to a loss-of-function mechanism for RASAL1 in BC, in concordance with previous data indicating that it might act as a tumor suppressor gene in thyroid cancer [24]. Moreover, germline RASAL1 variants may be relatively frequent in patients with apparently sporadic thyroid carcinoma with follicular features [25].

The 22 patients carrying ROS1 variants had a mean age at BC diagnosis of 41.36 years, while the 5 patients with RASAL1 variants had been diagnosed with BC at 36.2 years of age on average, compared to 38.29 years of patients with no variant detected in ROS1/RASAL1. More than half patients with ROS1 variants had a relevant family history, while RASAL1 carriers were mainly clinically sporadic early-onset cases. Features of patients are reported in Appendix A.

Sequencing data for ROS1 and RASAL1 genes were retrieved in silico from WES analysis of 197 Italian subjects with no reported cancer history, who were selected as controls. For collapsing analysis comparing gene frequencies between cases (131) and controls (197), we used stringent criteria as previously described by Povysil et al. [26], selecting only non-synonymous coding changes, such as missense, nonsense, frameshift, splice acceptor and donor variants, having a minor allele frequency (MAF) <0.001 in gnomAD database, both in BC cases and in control group. According to these guidelines, we removed three ROS1 variants (c.-32C > T, p.Ser79=, and p.Ser370Pro) and a synonymous RASAL1 variant (p.Ile84=) in the number of variants identified in the BC cases.

A significantly higher proportion of ROS1 non-synonymous variant carriers was observed among BC cases in comparison to controls (*p* = 0.0401; Fisher’s exact test, Table 4).

We also evaluated the somatic mutations in ROS1 and RASAL1 present in BC in COSMIC database (v.91). 4.53% mutated cases in ROS1 (217/4792) and 1.28% cases in RASAL1 (33/2583) were reported. Somatic mutations in BRCA1 were reported in 2.83% BC cases (229/8090), and in 4.06% in BRCA2 (306/7532). Comparing these frequencies, in RASAL1 a lower number of mutated cases was reported, compared to BRCA1 and BRCA2 genes (*p* < 0.00001 for each test; Chi-square test; Appendix A). Conversely, an increased number of ROS1-mutated BC cases, compared to BRCA1-mutated BC cases, was detected (*p* < 0.00001; Appendix A), further supporting a role for ROS1 in BC.

## 3. Discussion

Next-generation sequencing has made available the analysis of multiple genomic regions simultaneously, shortening the time and cost of gene tests, and high-throughput association studies have unveiled the presence of many independent predisposing BC loci, each contributing to a relatively small portion of familial BC risk [5,6,7,8]. However, integrating these data in the clinical management of patients present significant controversies, in particular for genes contributing moderate risks. Epidemiological studies have identified a number of modifiable risk factors (e.g., weight, use of hormones, alcohol consumption, physical activity, and breastfeeding, diet), each explaining a modest proportion of the variation in disease risk individually, but with a substantial effect on BC predisposition, when combined together and with unmodifiable factors (e.g., genes, menstrual and/or reproductive history, family history of BC, and prior benign breast disease) [27,28]. It has been shown that changing the risk factor profile (low weight gain, no alcohol consumption, high physical activity level, breastfeeding, and no hormone use) could reduce the rate of postmenopausal BC by more than 34% [29], and that the reduction in absolute risk achievable by changing modifiable risk factors is larger for those who are at higher risk from nonmodifiable factors [30]. In addition, women found to be at elevated risk because of unmodifiable factors (e.g., pathogenic gene variant carriers) could be more motivated to adopt healthier lifestyles to lower their risk of BC if they have a better understanding of the potential gains. Identifying the fraction of the preventable cases can also better tailor the risk factor modification actions only to high-risk cases. 

Moreover, several studies have identified specific molecular signatures with actionable targets in BC, generating a growing catalog of genomic mutations that facilitate tailored approaches to advanced BC, based on molecular analyses [31,32,33]. Therefore, an integrated molecular testing has to become an integral part of BC management. The use of whole exome sequencing in unexplained familial BC clusters may help identify novel of causal/predisposing variants in BC patients. In our study, we selected for the analysis affected first-degree cousins, who share 1/8 of their genome, therefore reducing the number of variants shared due to identity-by-descent. Under this assumption, we were able to identify novel or rare predicted pathogenic variants in *ROS1*, *RASAL1* and *POLN*, and *NPL* in the three cousin pairs included in the exome analysis. *NPL* encodes for N-acetylneuraminate pyruvate lyase, an enzyme regulating the cellular concentrations of N-acetyl-neuraminic acid (sialic acid) [34,35]. As reported in Human Protein Atlas, NPL overexpression seemed to correlate with poor survival in liver and renal cancers, but little is known regarding the corresponding altered molecular pathways. *POLN* encodes for the DNA Polymerase nu involved in cross-link repair, homologous recombination and binding to BRCA1 [36]. *ROS1* and *RASAL1* are two genes with established roles in tumor development and progression in different types of cancer [21,24,25,37,38].

Targeted gene screening allowed us to identify additional pathogenic variants in *RASAL1* and *ROS1*. In particular, we found a significant increase in the number of *ROS1*-mutated individuals in BC cases (131), compared to population controls (197 Italian individuals). Instead, the contribution of *NPL* and *POLN* to BC predisposition was not supported by the additional screening of cancer cases. In the 131 independent BC patients, we detected mainly *RASAL1* loss-of-function variants (Table 2), in concordance with a tumor suppressor role of RASAL1 in cancer [19,24,25]. RASAL1 stimulates the GTPase activity of normal RAS p21, but not its oncogenic counterpart. Acting as a suppressor of RAS function, the protein enhances the intrinsic GTPase activity of RAS proteins resulting in the inactive GDP-bound form of RAS, thereby controlling cellular proliferation and differentiation [19,20]. Recent data have shown that exogenous expression of RASAL1 in MCF-7 and MDA-MB-231 sensitized the response to hypoxia treatments, associated with its ability to directly reduce HIF-1α (Hypoxia-inducible factor 1-alpha) expression, inhibiting migration and decreasing the accumulation of reactive oxygen species [21]. Conversely, *RASAL1* knockdown reversed the cellular response to hypoxia [21]. Somatic driver gene point mutations in *RASAL1* in follicular thyroid cancer have also been reported [24]. Germline mutations in *RASAL1* have been identified in differentiated thyroid cancer (DTC) and in patients with Cowden Syndrome characterized by both BC and DTC features [25]. Intriguingly, in the family ITA_BC02 carrying the *RASAL1* variant there were also DTC cases. Therefore, we suggest that, similarly to other cancers characterized by *RASAL1* mutations, the loss of this tumor suppressor might promote BC development.

*ROS1* encodes for an orphan tyrosine kinase receptor and its somatic rearrangements leading to the fusion of the kinase domain to different genes have been described in sarcomas, glioblastoma multiforme and lung tumors (non-small cell lung cancer): for a review, see [39,40,41]. The natural ligand of ROS1 is still unknown, whereas the proteins derived by the genomic rearrangements present a plethora of different N-terminal domains from different genes fused in frame with the cytoplasmic portion of ROS1, including the tyrosine kinase [42]). These rearranged proteins are abnormally activated and stimulated unrestricted cell growth and proliferation, with the activation of several pathways, including SHP2/PI3K/Akt/mTOR pathways [43]. The *ROS1* rearrangements act as oncogenic drivers, and many tyrosine kinase inhibitors (TKIs) have been developed to target these driver oncogene products [44,45]. In lobular BC, the loss of E-cadherin expression that occurs early in the tumorigenic process [46], could be rescue by ROS1 synthetic lethality, an effect clinically actionable using ROS1 inhibitors [47]. However, many patients presenting tumors with somatic *ROS1* rearrangements develop a resistance to TKIs, due to the acquisition of secondary mutations in the ROS1 tyrosine kinase domain that are refractory to TKI inhibition and maintain the receptor abnormally activated [48,49]. The progress in developing new TKIs to overcome *ROS1* acquired secondary mutations is currently producing novel drugs for a targeted therapy [50,51]. 

Recently, *ROS1* pathogenic somatic variants were identified in inflammatory breast cancer [52] and *ROS1* somatic mutations are present in the COSMIC database. However, germline pathogenic variants have not been reported in hereditary BC. We characterized the splicing effect of the specific germline *ROS1* variant c.6005 + 2T > G identified in one of the cousin pairs analyzed by WES and in one unrelated BC patient undergone NGS target sequencing. We could establish that the variant induced the use of a cryptic splice site, generating a premature stop codon in the cytoplasmic tail of ROS1 receptor in the in-vitro cell model COS7 cells, widely used to study human splicing alterations, because they present a splicing apparatus comparable to the one in *H. sapiens* [22,53]. Although patient RNA is usually preferred for splicing analysis, several issues, such as availability, or degradation of aberrant transcripts through nonsense-mediated mRNA decay (NMD), may make it difficult to assess the effect of a variant allele on splicing. Minigene assays is an alternative method that show high sensitivity and specificity in the assessment of aberrant splicing caused by genetic sequence variants [54,55]. The unavailability of the corresponding breast cancer cells or pathological breast cancer tissues from the affected individuals carrying the constitutive germline variant c.6005 + 2T > G in *ROS1* prevented us from investigating whether the truncated protein was produced and could act with a dominant negative effect, shown for example by the truncated TrkB receptor [56] or promote an aberrant signal activation as shown by the truncated and constitutively active form of the EGF receptor, a key determinant of tumor growth and progression in cancer [57,58]. Interestingly, the majority of the other germline predicted pathogenic variants, found with a significant increase in the BC patient group, clustered to the cytoplasmic tail of ROS1, similarly to the variants conferring resistance to classic TKIs in *ROS1*-rearranged cancers and for which novel drugs are currently developed [50,51]. 

Our study provided the identification of novel and rare pathogenic constitutive variants in *ROS1* in hereditary BC, but the case study is relatively small; therefore, further studies in additional independent cohorts of familial BC cases is warranted to investigate the contribution of germline *ROS1* variants to breast cancer predisposition.

The discovery of additional genes predisposing to familial BC is crucial to identify the altered pathways in BC and is paramount in the development of novel agents that can target non-endocrine pathways, cellular proliferation and tumor progression. Moreover, identifying novel genetic factors conferring BC predisposition is crucial to predict who is at increased risk of developing the disease, allowing for early detection and therapy, and optimized patient management. 

## 4. Materials and Methods 

### 4.1. Patients

At the Cancer Genetics Clinic of the Hospital S.Orsola-Malpighi, BRCA testing is offered to patients with BC suspected to carry hereditary predisposition according to regional criteria [59]; about 18% of patients tested are found to carry a pathogenic variant and 6% carry variants of unknown significance; the remaining 76% test negative for *BRCA* variants [60]. Among kindreds highly suspicious for HBC with no *BRCA* variants found, we selected for WES analysis three families in which two 1st degree cousins had been tested. The pedigrees of the three families included in the Whole exome sequencing (WES) analysis are presented in Figure 1.

A consecutive series of 131 patients with familial and/or early-onset breast cancer who had undergone complete *BRCA1/2* analysis (sequencing and MLPA) with negative results were then enrolled in the study as confirmation set, upon informed consent. The study was approved by the Ethical Board of Hospital S.Orsola-Malpighi, Bologna, Italy (113/2013/O/Tess) and was in compliance with the Helsinki declaration. Features of the patients are reported in Appendix A.

### 4.2. WES Analysis

WES was performed on genomic DNA extracted from peripheral blood (QIAGEN mini kit) from three affected first-degree cousin-pairs, who tested negative for *BRCA1/2*. Dual-index paired-end libraries followed by exome enrichment were prepared starting from 100 ng genomic DNA, according to the Nextera Rapid Capture Enrichment protocol (Illumina, San Diego, CA USA). The captured regions were sequenced on the Illumina HiScanSQ platform for 200 cycles (100 cycles paired-ends; Illumina). The read files were aligned to hg19 version of the human genome sequencing, annotation and variant prioritization was performed according to our internal pipeline for exome annotation as previously reported [61]. The identified variants were confirmed by PCR and direct sequencing. Sequencing data for *RASAL1*, *ROS1* genes were retrieved in silico from 197 WES data of Italian subjects without reporting cancer history, who were selected as controls. Variants were called using GATK “HaplotypeCaller” and variant calls were recalibrated before being annotated with the Ensembl tool Variant Effect Predictor (VEP). The number of rare (MAF < 0.001) single nucleotide variants and small indels predicted to functionally impact the RASAL1 and ROS1 proteins, was obtained by custom scripts. Fisher’s exact test was used to test the null hypothesis of equality of proportions of subjects with at least one variant in these genes.

### 4.3. Splice Site Analysis

The genomic regions surrounding *ROS1* exon 37 and *NPL* exon 2 from the corresponding heterozygous carriers were amplified using primers 5′-GAATTCGTCTTGCTGGGAA-3′(*ROS1*) and 5′-GGGGCATTCGGACTTTGCCTGGGAGTTTT-3′ (*NPL*), inserting an *EcoRI* restriction site, and primers 5′-TCAGCAGACAAACTCCAAGCTCTAGA-3′ (*ROS1*), and 5′-CCTCTAGATTAGAAAGGCCAAGGCTGTG-3′ (*NPL*) inserting a *XbaI* restriction site, respectively. We generated the mini-gene reporter as previously described [22]. The PCR products were cloned into the digested P1-pAltermax and plasmids sequenced in order to identify the plasmids with the wild-type (wt) or the variant alleles. The splicing alteration analysis was carried out in COS7 cells (derived from monkey kidney tissue) grown in DMEM, 10% fetal bovine serum, 2 mM L-glutamine, 100 U/ mL penicillin and 100 μg/mL streptomycin (Sigma-Aldrich; St Louis, MO, USA), in a humidified incubator at 37 °C with 5% CO_2_. 3 × 10^5^ COS7 cells were plated in six-well plate for transfection using Lipofectamine reagent, according to the manufacturer’s instructions (Lipofectamine 2000, Thermo Fisher Scientific, Waltham, MA, USA). Total RNA was extracted after 48 h from transfection using the RNeasy kit (QIAGEN, Hilden, Germany) and subjected to *DNase I* digestion (Fermentas, Thermo Fisher Scientific) for 30 min at 37 °C. 1 μg of *DNase I*–treated RNA was used for reverse transcription with random hexamers and the Multiscribe RT system (Thermo Fisher Scientific) at 48 °C for 40 min in a final volume of 50 μL. RT-PCR was performed with primers specific for the P1-pAltermax vector, forward T7EE 5′-AAGGCTAGAGTACTTAATACGA-3′ and reverse PMaxR 5′-TATCATGTCTGCTCGAAGCATTA-3′. RT-PCR was carried out in the following conditions: 2 μL cDNA, 2.5 mM MgCl_2_, 0.5 mM dNTPs, 0.5 μM primers, 5% dimethyl sulfoxide in a final volume of 20 μL using the 2× KAPA Fast Taq Polymerase Master mix (KAPA Biosystems, Sigma-Aldrich). The RT-PCR products were run on 2.5% *w*/*v* agarose gel in TBE 1× buffer and visualized under UV light using ethidium bromide (Sigma-Aldrich).

### 4.4. Target Gene Screening

The Illumina TruSeq Custom Amplicon Kit was used to capture all exons and the flanking sequences of *NPL*, *POLN*, *RASAL1* and *ROS1* genes. For all genes, custom oligos specific to our targeted regions of interest were designed using DesignStudio [23] and amplicon length averaged 250 base pairs (2 × 150 base pairs reads length in paired-end mode). The number of amplicons per gene in the panel varies from 15 to 70, with a total of 143 amplicons and a mean coverage of 200×. Genomic DNA from peripheral blood was extracted using QIAGEN DNA mini kit (QIAGEN). The TruSeq Custom Amplicon sequencing assay was performed according to the manufacturer’s protocol (Illumina). All DNA samples were diluted to 10 ng/L and then were hybridized to the custom oligo pool (CAT). Upstream and downstream oligos were extended and ligated. Then, the result products containing the targeted regions of interest were amplified. Sample normalization of each library was performed according to Illumina bead-based normalization manufacturer’s protocol for balanced representation in pooled libraries. Pooled Amplicon Library (PAL) preparation was performed by combining 5 μL of each normalized library in one tube. PAL was quantified by Qubit dsDNA HS Assay kit (ThermoFisher Scientific) and diluted to a final concentration of 10 pM to be loaded on the flow cell. Runs were performed on Illumina MiSeq sequencer with a V2 flow cell (300 cycles). Bioinformatics analysis (including demultiplexing, reads alignment and variant calling) was performed using the MiSeq provided software (Illumina). Variant annotation and prioritization were performed using the dedicated cloud-based software BaseSpace, using the Variant Studio program (Illumina).

## 5. Conclusions

In our study, we focused on the identification of germline variants in familial BC, through whole exome and target NGS sequencing. Considering the relatively small number of familial BC individuals analyzed, the evaluation of *ROS1* and *RASAL1* as potential predisposing genes is warranted through further screening in independent patient groups. Identifying novel genetic factors conferring BC predisposition is crucial to predict who is at increased risk of developing the disease, allowing for early detection and therapy, and optimized patient management. 

## Figures and Tables

**Figure 1 cancers-12-02539-f001:**
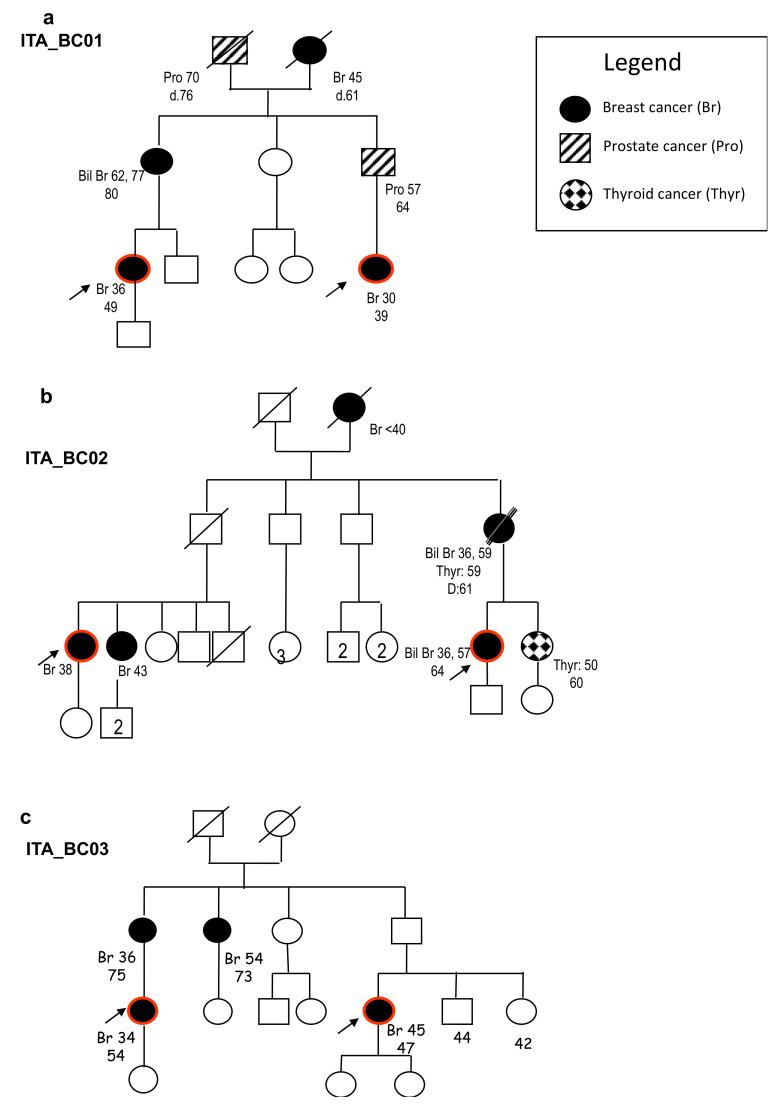
Family trees of the families included in the whole exome sequencing (WES) analysis. We reported in the legend the different types of tumors identified in the corresponding individuals in the different generations. Red circles indicate the subjects analyzed via WES. (**a**) Family tree of the first-degree cousins carrying the *ROS1* (ROS Proto-Oncogene 1, Receptor Tyrosine Kinase) splice variant. (**b**) Family tree of the first-degree cousins carrying the *RASAL1* (RAS Protein Activator Like 1) and *POLN* (DNA Polymerase Nu) missense variants. (**c**) Family tree of the first-degree cousin carrying the *NPL* (N-Acetylneuraminate Pyruvate Lyase) splice variant.

**Figure 2 cancers-12-02539-f002:**
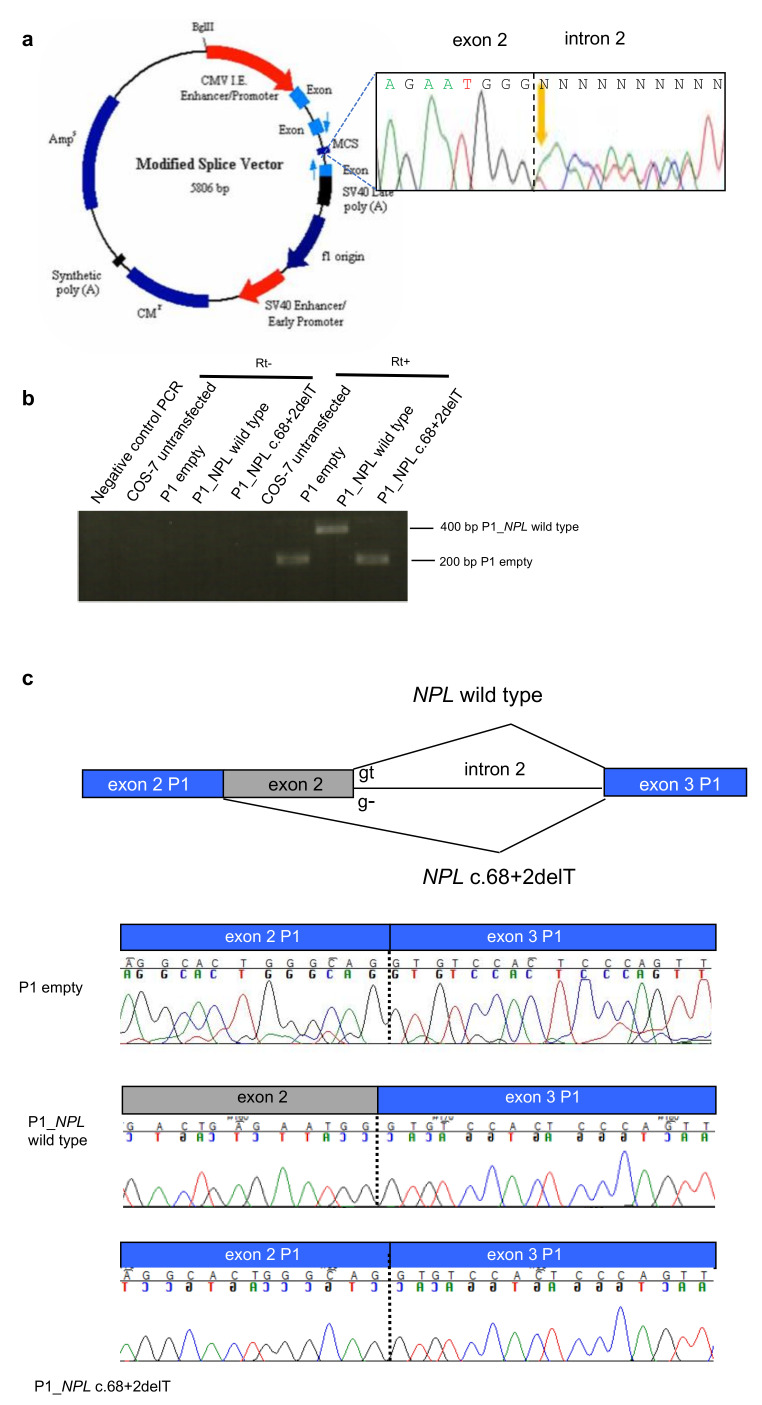
The NPL c.68 + 2delT mutation alters the splicing. (**a**) Map of the minigene system. In the MCS (multiple cloning site) the genomic region surrounding the mutation was inserted starting from the PCR product of one heterozygous patient (electropherogram in black box). (**b**) Reverse Transcriptase-PCR (RT-PCR) from RNA extracted from COS7 cells transfected with the different plasmids: empty, carrying NPL wild type, carrying the NPL variant. (**c**) Electropherograms of the RT-PCR products showing that wild type NPL produced a correct splicing, whereas the mutant NPL induced exon 2 skipping.

**Figure 3 cancers-12-02539-f003:**
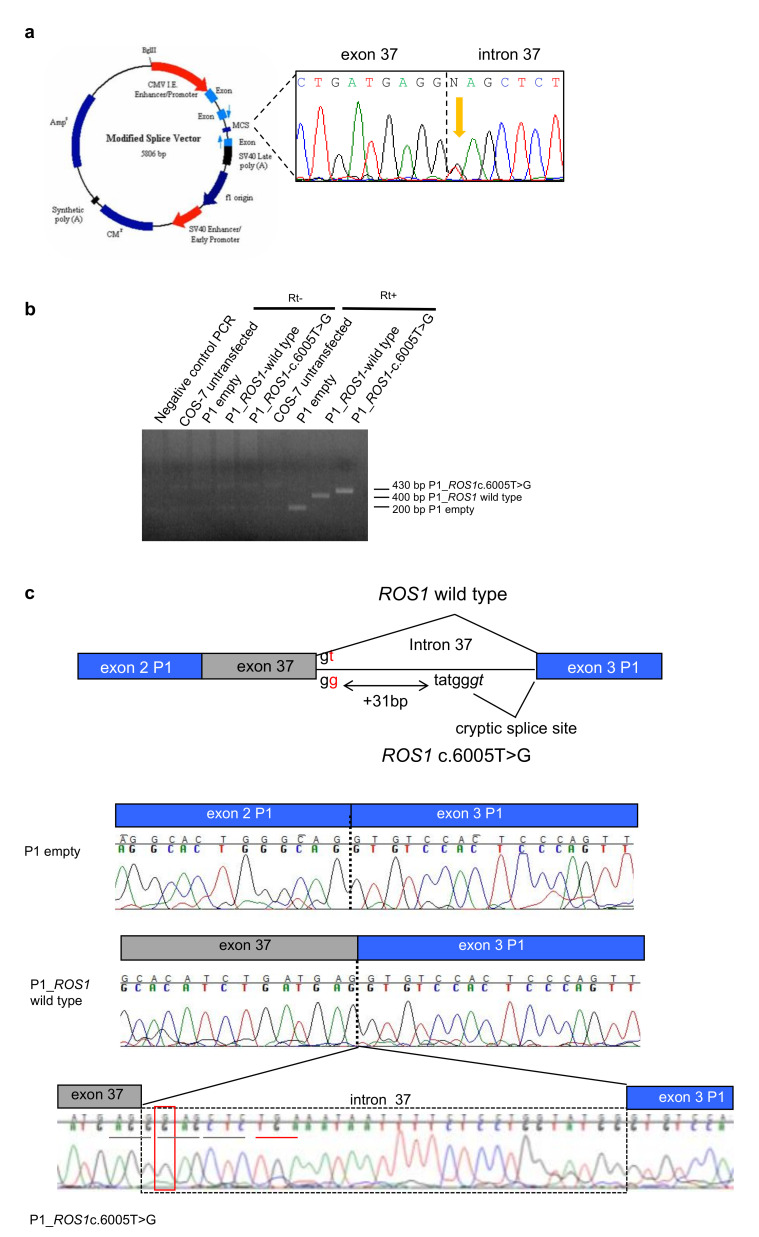
The ROS1 c.6005 + 2T > G variant alters the splicing. (**a**) Map of the minigene system. In the MCS the genomic region surrounding the mutation was inserted starting from the PCR product of one heterozygous patient (electropherogram in black box). (**b**) RT-PCR of cDNA from COS7 cells transfected with the different plasmids: empty, carrying ROS1 wild type; carrying the ROS1 variant. (**c**) Electropherograms of the RT-PCR products, showing that wild type ROS1 allele produces a correct splicing, whereas in the variant one a cryptic splice site in the intron 37 is used, generating a longer transcript with the insertion of a premature stop codon in the open reading frame.

**Table 1 cancers-12-02539-t001:** Single nucleotide variants/indels identified by WES in the three families with hereditary breast cancer.

Family ID	Genomic Coordinates (hg19)	Gene	Functional Effect	dbSNP	PolyPhen-2 Scores	M-CAP Score	MAF gnomAD (European) ^a^
ITA_BC01	6: g.117639349A > C	*ROS1* (NM_002944)	c.6005 + 2T > G	rs200482357	--	--	0.000008850
ITA_BC02	12: g.113543542A > G	*RASAL1* (NM_001193521)	p.Phe602Leu	rs142556970 ^b^	HumDiv = 0.136 (benign) HumVar = 0.217 (benign)	0.207 (Possibly pathogenic)	0.002692452
ITA_BC02	4: g.2209742T > G	POLN (NM_181808)	p.Asp229Ala	rs747058815	HumDiv = 1.000 (Probably damaging) HumVar = 0.993 (Probably damaging)	0.032 (Possibly pathogenic)	3.53638 × 10^−5^
ITA_BC03	1:g.182763575GT > G	NPL (NM_030769)	c.68 + 2delT	rs755026414	--	--	0.000425763

(a) Allele frequency in European non-Finnish population, as accessed in gnomAD v2.1. [12].

**Table 2 cancers-12-02539-t002:** Rare/novel single nucleotide variants in the *ROS1* gene, identified in breast cancer (BC) patients via next-generation sequencing (NGS) target analysis.

Patient ID	chr6 (hg19)	cDNA NM_002944	Functional Effect	dbSNP	PolyPhen-2 Scores	M-CAP Score	MAF gnomAD (European) ^a^
BC14623 BC12674	g.117746850C > T	c.-32C > T	5′UTR	rs376700858	--	--	0/127478 (*p* < 0.00001) ^b^
BC12132 ^d^ BC11298 BC15651	g.117730797C > T	c.237G > A	p.Ser79= (removing ESE)	rs55736087	--	--	0.004862 628/129166 (*p* = 0.1371) ^b^
BC10557 BC10536	g.117715381A > G	c.1108T > C	p.Ser370Pro	rs56274823	HumDiv = 0.917 Possibly damaging HumVar = 0.446 Benign	0.086 Possibly pathogenic	0.003072 390/126944 (*p* = 0.1937) ^b^
BC9762	g.117710875C > T	c.1397G > A	p.Arg466Gln	rs140178288	HumDiv = 0.957 Probably damaging HumVar = 0.246 Benign	0.048 Possibly pathogenic	0.000007752 1/129000 (*p* = 0.004) ^b^
BC9631	g.117710828G > T	c.1444C > A	p.Leu482Met	rs757273336	HumDiv = 1.000 Probably damaging HumVar = 0.998 Probably damaging	0.075 Possibly pathogenic	0/113636 (*p* = 0.0023) ^b^
BC14139	g.117710578G > A	c.1694C > T	p.Ser565Leu	rs142303126	HumDiv = 0.830 Possibly damaging HumVar = 0.124 Benign	0.042 Possibly pathogenic	0.0005425 70/129034 (*p* = 0.1342) ^b^
BC11084	g.117709072T > C	c.1885A > G	p.Ile629Val (inserting splice donor site)	rs142877218	HumDiv = 0.000 Benign HumVar = 0.001 Benign	0.027 Possibly pathogenic	0.00009297 12/129076 (*p* = 0.026) ^b^
BC13285	g.117704600C > G	c.2376G > C	p.Met792Ile (removing ESE)	rs140511382	HumDiv = 0.279 Benign HumVar = 0.081 Benign	--	0.00002323 3/129128 (*p* = 0.0081) ^b^
BC11925	g.117704565G > T	c.2411C > A	p.Thr804Asn	rs200615700	HumDiv = 0.999 Probably damaging HumVar = 0.922 Probably damaging	0.085 Possibly pathogenic	0.0003331 43/129096 (*p* = 0.0854) ^b^
BC12188	g.117678020T > A	c.3913A > T	p.Ile1305Phe	--	HumDiv = 0.553 Possibly damaging HumVar = 0.110 Benign	0.058 Possibly pathogenic	Novel
BC14203 BC17333 BC18206	g.117642495C > T	c.5704G > A	p.Glu1902Lys	rs9489124	HumDiv = 0.923 Probably damaging HumVar = 0.122 Benign	--	0.0009854 127/128884 (*p* = 0.0024) ^b^
BC14840	g.117642468C > G	c.5731G > C	p.Gly1911Arg	--	HumDiv = 0.997 Probably damaging HumVar = 0.925 Benign	0.033 Possibly pathogenic	Novel
BC14177 ^c^	g.117639349A > C	c.6005 + 2T > G	Altered donor splice site	rs200482357	--	--	0.000008850 1/112990 (*p* = 0.0046) ^b^
BC12132 ^d^	g.117622137C > T	c.6733G > A	p.Gly2245Ser	rs142264513	HumDiv = 0.196 Benign HumVar = 0.063 Benign	0.032 Possibly pathogenic	0.001065 124/116398 (*p* = 0.2451) ^b^
BC13141 BC13389 BC16373	g.117631301C > T	c.6377G > A	p.Arg2126Gln	rs199882276	HumDiv = 1.000 Probably damaging HumVar = 0.998 Probably damaging	0.096 Possibly pathogenic	0.0006670 86/128930 (*p* = 0.0008) ^b^

(a) Allele frequency in European non-Finnish population, as accessed in gnomAD v2.1. [12]. In the second row, we reported the number of variant alleles vs the total alleles for the specific position in the database. (b) *p* values reported for Fisher’s exact test, calculated considering the number of variant alleles for the BC cases (*n* = 131 cases analyzed; 262 total alleles), and the frequency of the variant alleles as reported in gnomAD v2.1. [12]. Significant *p* values are indicated in italics. (c) The splice variant identified in individual BC14177 is the same identified in the unrelated family ITA_BC01 via WES analysis. (d) Patient BC12132 carried two different variants in *ROS1*. Abbreviations as reported for Table 1.

**Table 3 cancers-12-02539-t003:** Rare/novel single nucleotide variants in the *RASAL1* gene, identified in BC patients via target NGS analysis.

Patient ID	chr.12 (hg19)	cDNA NM_001193520	Functional Effect	dbSNP	PolyPhen-2 Scores	M-CAP Score	MAF gnomAD (European) ^a^
BC14290	g.113128174G > A	c.127C > T	p.Ala43Thr	--	HumDiv = 0.986 Probably damaging HumVar = 0.969 Probably damaging	--	Novel
BC14401	(1) g.113565948C > T (2) g.113127862T > A	(1) c.158G > A (2) c.248A > T	(1) p.Trp53Ter (2) p.Ile83Asn	rs139329607	(2) HumDiv = 0.880 Possibly damaging HumVar = 0.718 Possibly damaging	--	(1) 0.000008797 1/113678 (*p* = 0.0046) ^b^ (2) Novel
BC17895	g.113565663G > A	c.252C > T	p.Ile84=	rs61759864	--	--	0.001356 175/129046 (*p* = 0.3004) ^b^
BC18101	g.113546005C > T	c.1400G > A	p.Ser466Asn	--	HumDiv = 0.494 Possibly damaging HumVar = 0.474 Possibly damaging	0.099 Possibly pathogenic	Novel
BC16851	g.113553568 delA	c.875 delT	p.Leu292CysfsTer5	--	--	--	Novel

(a) Allele frequency in European non-Finnish population, as accessed in gnomAD v2.1. [12]. In the second row, we reported the number of variant alleles vs the total alleles for the specific position in the database. (b) *p* values reported for Fisher’s exact test, calculated considering the number of variant alleles for the BC cases (*n* = 131 cases analyzed; 262 total alleles), and the frequency of the variant alleles as reported in gnomAD v2.1. [12]. Significant *p* values are indicated in italics.

**Table 4 cancers-12-02539-t004:** Contingency table of mutated subjects in *ROS1* and *RASAL1* genes, identified among cases and controls.

Individuals	*ROS1*			*RASAL1*		
	Cases	Controls	Total	Cases	Controls	Total
Mutated	16	11	27	4	4	8
Not mutated	115	186	301	127	193	320
Total	131	197	328	131	197	328
	*p* = 0.0401 ^a^			*p* = 0.7179 ^a^		

(a) Fisher’s exact test was used to test the null hypothesis of equality of proportions of subjects with at least one variant in these genes.

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
