# Peer review of "RASAL1 and ROS1 Gene Variants in Hereditary Breast Cancer"

_cancers, 2020, doi:10.3390/cancers12092539_

Round 1

Reviewer 1 Report

The paper reports results of an interesting study conducted in order to identify candidate gene variants for hereditary breast cancer. The manuscript is original and generally well written, clearly presented, and the study design is appropriate.

I suggest to improve the abstract adding the aim of the study. The discussion on possible risk factors, modifiable and unmodifiable, could be deepened. Furthermore, I think that could be interesting to identify and discuss possible differences among cases and controls in terms of age, distribution of other risk factors, etc.. In the Discussion section strengthens and limits of the study should be appropriately discussed by the Authors. Possible clinical and public health implication of results should be further discussed, also considering other published results (References should be added).

Author Response

Dear Editors and Reviewers,

we would like to thank you all for your help and collaboration in considering our manuscript for publication in Cancers. We think that all received comments and suggestions helped us in improving our manuscript.

Please find as follows the replies to the Comments.

# Reviewer 1

Comments and Suggestions for Authors

The paper reports results of an interesting study conducted in order to identify candidate gene variants for hereditary breast cancer. The manuscript is original and generally well written, clearly presented, and the study design is appropriate.

I suggest to improve the abstract adding the aim of the study.

We thank Reviewer 1 for the important suggestion. We modified the Abstract as follows: page 1, lines 23-24: “... We aimed to identify new predisposing variants in familial BC patients using next-generation sequencing approaches”.

The discussion on possible risk factors, modifiable and unmodifiable, could be deepened. Furthermore, I think that could be interesting to identify and discuss possible differences among cases and controls in terms of age, distribution of other risk factors, etc.. In the Discussion section strengthens and limits of the study should be appropriately discussed by the Authors. Possible clinical and public health implication of results should be further discussed, also considering other published results (References should be added).

We thank the Reviewer 1 for these insightful suggestions, which were taken into account by evaluating several epidemiological and clinical features (including sex, age, familial history) of the patients showing ROS1 and RASAL1 variants, although the number of cases was relatively too small to achieve statistical significance. These data are now inserted in the novel Supplementary Table 3 and discussed in the results and discussion sections.

We added in the Results the following paragraph: page 16, lines 220-224, “The 22 patients carrying ROS1 variants had a mean age at BC diagnosis of 41.36 years, while the five patients with RASAL1 variants had been diagnosed with BC at 36.2 years of age on average, compared to 38.29 years of patients with no variant detected in ROS1/RASAL1. More than half patients with ROS1 variants had a relevant family history, while RASAL1 carriers were mainly clinically sporadic early-onset cases. Features of patients are reported in Supplementary Table 3.”

In the Discussion section we modified the text and inserted the corresponding new references as follows: page 17, lines 255-267, “Epidemiological studies have identified a number of modifiable risk factors, (e.g. weight, use of hormones, alcohol consumption, physical activity, breastfeeding, diet) each explaining a modest proportion of the variation in disease risk individually, but with a substantial effect on BC predisposition, when combined together and with unmodifiable factors (e.g. genes, menstrual and/or reproductive history, family history of BC, prior benign breast disease) [19,20]. It has been shown that changing the risk factor profile (low weight gain, no alcohol consumption, high physical activity level, breastfeeding, and no hormone use) could reduce the rate of postmenopausal BC by more than 34% [21], and that the reduction in absolute risk achievable by changing modifiable risk factors is larger for those who are at higher risk from nonmodifiable factors [22]. In addition, women found to be at elevated risk because of unmodifiable factors (e.g. pathogenic gene variant carriers) could be more motivated to adopt healthier lifestyles to lower their risk of BC if they have a better understanding of the potential gains. Identifying the fraction of the preventable cases can also better tailor the risk factor modification actions only to high-risk cases.”

Page 18, lines 340-343, “Our study provided the identification of novel and rare pathogenic constitutive variants in ROS1 in hereditary BC, but the case study is relatively small, therefore further studies in additional independent cohorts of familial BC cases is warranted to investigate the contribution of germline ROS1 variants to breast cancer predisposition.”

Page 18, lines 346-348, “. Moreover, identifying novel genetic factors conferring BC predisposition is crucial to predict who is at increased risk of developing the disease, allowing for early detection and therapy, and optimized patient management.

Reviewer 2 Report

Although the results and conclusion of this paper are novel and important, but the manuscript lack to elucidate directed or possible mechanisms. The authors try to develop a risk prediction model based on data bank. At least, some concerns should be addressed:

1) As the human protein atlas bank indicates RASAL1 expression high in ovarian cancer, breast and thyroid tumors. Not only for BC, especially authors want to emphasize RASAL1 and ROS1 gene variants in hereditary BC negative at the BRCA1/2 testing. Just for discussion part of this study illustrated these databases in the clinical management of patients present significant controversies, in particular for genes contributing moderate risks. Many publications used analysis many databases to predict prevalence for disease are rapidly and economically. Although the authors used cos-7 cells to investigate possible splice site alterations in NPL and ROS1 on the final transcripts. We hope the authors should elucidate the possible splice site with human familial “breast cancer cells” or “tissue”, at least, cell line evidence is required.

2) Some publications indicate importance of ROS1 major in cancer development (e.g. NSCLC, gastric cancer, ovarian cancer, cholangiocarcinoma, colorectal cancer), not in susceptibility of breast cancers. So, I suggest that the authors can provide paragraphs in discussion part for predicting RASAL1/ ROS1 compared with predicting BRCA1/2 in hereditary BC.

Author Response

Dear Editors and Reviewers,

we would like to thank you all for your help and collaboration in considering our manuscript for publication in Cancers. We think that all received comments and suggestions helped us in improving our manuscript.

Please find as follows the replies to the Comments.

# Reviewer 1

Comments and Suggestions for Authors

The paper reports results of an interesting study conducted in order to identify candidate gene variants for hereditary breast cancer. The manuscript is original and generally well written, clearly presented, and the study design is appropriate.

I suggest to improve the abstract adding the aim of the study.

We thank Reviewer 1 for the important suggestion. We modified the Abstract as follows: page 1, lines 23-24: “... We aimed to identify new predisposing variants in familial BC patients using next-generation sequencing approaches”.

The discussion on possible risk factors, modifiable and unmodifiable, could be deepened. Furthermore, I think that could be interesting to identify and discuss possible differences among cases and controls in terms of age, distribution of other risk factors, etc.. In the Discussion section strengthens and limits of the study should be appropriately discussed by the Authors. Possible clinical and public health implication of results should be further discussed, also considering other published results (References should be added).

We thank the Reviewer 1 for these insightful suggestions, which were taken into account by evaluating several epidemiological and clinical features (including sex, age, familial history) of the patients showing ROS1 and RASAL1 variants, although the number of cases was relatively too small to achieve statistical significance. These data are now inserted in the novel Supplementary Table 3 and discussed in the results and discussion sections.

We added in the Results the following paragraph: page 16, lines 220-224, “The 22 patients carrying ROS1 variants had a mean age at BC diagnosis of 41.36 years, while the five patients with RASAL1 variants had been diagnosed with BC at 36.2 years of age on average, compared to 38.29 years of patients with no variant detected in ROS1/RASAL1. More than half patients with ROS1 variants had a relevant family history, while RASAL1 carriers were mainly clinically sporadic early-onset cases. Features of patients are reported in Supplementary Table 3.”

In the Discussion section we modified the text and inserted the corresponding new references as follows: page 17, lines 255-267, “Epidemiological studies have identified a number of modifiable risk factors, (e.g. weight, use of hormones, alcohol consumption, physical activity, breastfeeding, diet) each explaining a modest proportion of the variation in disease risk individually, but with a substantial effect on BC predisposition, when combined together and with unmodifiable factors (e.g. genes, menstrual and/or reproductive history, family history of BC, prior benign breast disease) [19,20]. It has been shown that changing the risk factor profile (low weight gain, no alcohol consumption, high physical activity level, breastfeeding, and no hormone use) could reduce the rate of postmenopausal BC by more than 34% [21], and that the reduction in absolute risk achievable by changing modifiable risk factors is larger for those who are at higher risk from nonmodifiable factors [22]. In addition, women found to be at elevated risk because of unmodifiable factors (e.g. pathogenic gene variant carriers) could be more motivated to adopt healthier lifestyles to lower their risk of BC if they have a better understanding of the potential gains. Identifying the fraction of the preventable cases can also better tailor the risk factor modification actions only to high-risk cases.”

Page 18, lines 340-343, “Our study provided the identification of novel and rare pathogenic constitutive variants in ROS1 in hereditary BC, but the case study is relatively small, therefore further studies in additional independent cohorts of familial BC cases is warranted to investigate the contribution of germline ROS1 variants to breast cancer predisposition.”

Page 18, lines 346-348, “. Moreover, identifying novel genetic factors conferring BC predisposition is crucial to predict who is at increased risk of developing the disease, allowing for early detection and therapy, and optimized patient management.

# Reviewer 2

Comments and Suggestions for Authors

Although the results and conclusion of this paper are novel and important, but the manuscript lack to elucidate directed or possible mechanisms. The authors try to develop a risk prediction model based on data bank. At least, some concerns should be addressed:

We thank the Reviewer 2 for the comment, however we would like to point out that we could not provide a risk prediction model based on data bank, but provided the data regarding the identifion of novel and rare predicted pathogenic variants in familial BC. Our sample size is relatively small to reach the power of developing any predictive model. We only compared the frequency of our variants in BC cases with a large control database usually used in human genetic studies (i.e. gnomAD) and our internal (i.e. genetic data of our Unit of medical genetics) database of Italian individuals with no reported history of cancer.

  • As the human protein atlas bank indicates RASAL1 expression high in ovarian cancer, breast and thyroid tumors. Not only for BC, especially authors want to emphasize RASAL1 and ROS1 gene variants in hereditary BC negative at the BRCA1/2 testing. Just for discussion part of this study illustrated these databases in the clinical management of patients present significant controversies, in particular for genes contributing moderate risks. Many publications used analysis many databases to predict prevalence for disease are rapidly and economically.

We reported the expression data in normal and cancer tissues of the genes where we identified germline variants (DNA from peripheral blood) in publicly available databases, to see the general expression of these genes, and compared to the general expression of two known BC genes such as BRCA1 and BRCA2, but the expression data are not related to the tissues of our patients, which we do not possess.

As mentioned in the response of reviewer 1, we added the features of the patients, considering the group with variants in RASAL1, the group with variants in ROS1 and the one with no variants as Supplementary table 3, and added in the Discussion literature data on modifiable and unmodifiable risk factors. We would like to point out that we analyzed a small group of BC cases and focused our attention on novel or very rare predicted pathogenic variants, therefore our study is not an association study, where generally anonymous SNPs are genotyped in large collections of cases and controls.

Although the authors used cos-7 cells to investigate possible splice site alterations in NPL and ROS1 on the final transcripts. We hope the authors should elucidate the possible splice site with human familial “breast cancer cells” or “tissue”, at least, cell line evidence is required.

We agree with the reviewer, but, as mentioned in the manuscript, we did not have any RNA or cell line derived from the specific patients carrying the constitutive variants in ROS1 and NPL, but only genomic DNA from blood was available. However, the minigene approach is an extensively used system to test specific alleles on splicing, as documented by several previously accepted studies.

We modified in concordance the Discussion as follows: Discussion, page 18, lines 320-331, “We characterized the splicing effect of the specific germline ROS1 variant c.6005+2T>G identified in one of the cousin pairs analyzed by WES and in one unrelated BC patient undergone NGS target sequencing.. We could establish that the variant induced the use of a cryptic splice site, generating a premature stop codon in the cytoplasmic tail of ROS1 receptor in the in-vitro cell model COS7 cells, widely used to study human splicing alterations, because they present a splicing apparatus comparable to the one in H.sapiens [15,45]. Although patient RNA is usually preferred for splicing analysis, several issues such as availability, or degradation of aberrant transcripts through nonsense-mediated mRNA decay (NMD), may make it difficult to assess the effect of a variant allele on splicing. Minigene assays is an alternative method that show high sensitivity and specificity in the assessment of aberrant splicing caused by genetic sequence variants [46,47]. The unavailability of the corresponding breast cancer cells or pathological breast cancer tissues from the affected individuals carrying the constitutive germline variant c.6005+2T>G in ROS1…”

2) Some publications indicate importance of ROS1 major in cancer development (e.g. NSCLC, gastric cancer, ovarian cancer, cholangiocarcinoma, colorectal cancer), not in susceptibility of breast cancers. So, I suggest that the authors can provide paragraphs in discussion part for predicting RASAL1/ ROS1 compared with predicting BRCA1/2 in hereditary BC.

We included a paragraph for RASAL1, since germline mutations in this gene have been reported in patients with Cowden syndrome, characterized by thyroid and/or breast cancers. Discussion, page 17, lines 299-301: “Therefore, we suggest that, similarly to other cancers characterized by RASALl mutations, the loss of this tumor suppressor might promote BC development.”

 Regarding ROS1, we report for the first time (at least to our knowledge) the detection of predicted pathogenic germline variants in breast cancer cases. The importance of ROS1 in cancer derives mostly from somatic rearrangements found in different types of tumors, which lead to abnormal receptor activation, but these are somatic rearrangements not germline (constitutive) changes. We tried to clarify these issues and indicated the effect of somatic ROS1 rearrangements as follows:

Discussion, page 18, lines 302-317: “ROS1 encodes for an orphan tyrosine kinase receptor and its somatic rearrangements leading to the fusion of the kinase domain to different genes have been described in sarcomas, glioblastoma multiforme and lung tumors (non-small cell lung cancer) [for a review see 31–33]. The natural ligand of ROS1 is still unknown, whereas the proteins derived by the genomic rearrangements present a plethora of different N-terminal domains from different genes fused in frame with the cytoplasmic portion of ROS1, including the tyrosine kinase [34]). These rearranged proteins are abnormally activated and stimulated unrestricted cell growth and proliferation, with the activation of several pathways, including SHP2 / PI3K / Akt / mTOR pathways [35]. The ROS1 rearrangements act as oncogenic drivers, and many tyrosine kinase inhibitors (TKIs) have been developed to target these driver oncogene products [36, 37]. In lobular BC, the loss of E-cadherin expression that occurs early in the tumorigenic process [38], could be rescue by ROS1 synthetic lethality, an effect clinically actionable using ROS1 inhibitors [39]. However, many patients presenting tumors with somatic ROS1 rearrangements develop a resistance to TKIs, due to the acquisition of secondary mutations in the ROS1 tyrosine kinase domain that are refractory to TKI inhibition and maintain the receptor abnormally activated [40, 41].”

We also suggest that further studies in additional independent cohorts of familial BC cases is warranted to investigate the contribution of germline ROS1 variants to breast cancer predisposition (page 18, lines 341-343). Any prediction of RASAL1/ROS1 compared to BRCA1/2 would be too preliminary, in absence of independent replications of our study, carried out on a small group of (well-characterized) BC cases. Nevertheless, we think it is important to consider variants predicted pathogenic in these genes, considering that (page 18, lines 346-348) identifying novel genetic factors conferring BC predisposition is crucial to predict who is at increased risk of developing the disease, allowing for early detection and therapy, and optimized patient management.

Round 2

Reviewer 2 Report

Although I mostly agree with the author's revision in this manuscript and response. But consideration for quality of Cancers journal, I still suggest the author's important finding can be applied in human familial “breast cancer cells”  at least.